# Prediction of ADMET Properties of Anti-Breast Cancer Compounds Using Three Machine Learning Algorithms

**DOI:** 10.3390/molecules28052326

**Published:** 2023-03-02

**Authors:** Xinkang Li, Lijun Tang, Zeying Li, Dian Qiu, Zhuoling Yang, Baoqiong Li

**Affiliations:** School of Biotechnology and Health Sciences, Wuyi University, Dongcheng Village, Jiangmen 529020, China

**Keywords:** ADMET, classification, machine learning, PLS-DA, AdaBoost, LGBM

## Abstract

In recent years, machine learning methods have been applied successfully in many fields. In this paper, three machine learning algorithms, including partial least squares-discriminant analysis (PLS-DA), adaptive boosting (AdaBoost), and light gradient boosting machine (LGBM), were applied to establish models for predicting the Absorption, Distribution, Metabolism, Excretion, and Toxicity (ADMET for short) properties, namely Caco-2, CYP3A4, hERG, HOB, MN of anti-breast cancer compounds. To the best of our knowledge, the LGBM algorithm was applied to classify the ADMET property of anti-breast cancer compounds for the first time. We evaluated the established models in the prediction set using accuracy, precision, recall, and F1-score. Compared with the performance of the models established using the three algorithms, the LGBM yielded most satisfactory results (accuracy > 0.87, precision > 0.72, recall > 0.73, and F1-score > 0.73). According to the obtained results, it can be inferred that LGBM can establish reliable models to predict the molecular ADMET properties and provide a useful tool for virtual screening and drug design researchers.

## 1. Introduction

Globally, cancer is the leading cause of death and an important barrier to improving life expectancy [1]. As the “pink killer” of women, breast cancer ranks first in the incidence of cancer in women worldwide and is one of the most common female malignant tumors. In 2020, the incidence of breast cancer in Chinese women was 59.0/100,000, ranking first in the incidence spectrum of female malignant tumors in China. In 2020, the mortality rate of breast cancer in Chinese women was 16.6/100,000, ranking fourth on the death spectrum of female malignant tumors in China [2,3]. The estrogen receptor alpha (ERα) has been considered a vital target for the treatment of breast cancer in recent years [4]. Based on related targets of breast cancer and estrogen signaling pathways, scientists are working hard to develop novel, high-efficiency, and low-toxic anti-breast cancer drugs.

ADMET (Absorption, Distribution, Metabolism, Excretion, and Toxicity) are the important properties of candidate drug molecules and are a key indicator in evaluating whether molecular compounds can be used as drugs. The experimental evaluation of ADMET properties costs significant labor power and material resources, and the workload cannot meet the demands of drug screening and lead optimization [5,6]. Fortunately, the ADMET properties can be theoretically predicted from the chemical structures using data mining approaches so that a large number of compounds can be evaluated prior to being obtained and assayed, thus minimizing failures in the process of drug research and development. The theoretical predictions of ADMET properties have been proven to be efficient in recent years [7] and are well established as a reliable and cost-effective approach to assist drug discovery [8].

Machine learning methods have shown their advantages in characterizing the hidden patterns of data and are thus employed to solve complex tasks [9] and have attracted wide attention in the fields of medicinal chemistry [10]. Researchers worldwide are continuing to advance ADMET properties prediction approaches, developing multi-function software and taking advantage of machine learning methods. To date, many ADMET prediction models have been developed using machine learning methods, such as multiple linear regression (MLR), principle component regression analysis (PCR), partial least squares (PLS), ridge regression (RR), and so on [11]. Sometimes, however, the relationship between the physicochemical properties of a compound and its biological activity may be too complex to be described by a linear function. Thus, more complex nonlinear machine learning methods are proposed and applied to capture more complex relationships between structure and activity, such as artificial neural networks (ANN), support-vector machine regression (SVM), random forest (RF), Naïve Bayes (NB), and so on [12,13]. In recent research, deep learning technology, such as deep neural networks (DNN), recurrent neural networks (RNN), convolutional deep neural networks (CNN), and their related implementation, has exhibited advantages in medicinal chemistry [14,15]. Moreover, there are several computational tools for predicting ADMET properties, such as ADMETlab [16], ADMETlab 2.0 [17], Pharmaco Kinetics Knowledge Base (PKKB) [7], ADMET structure-activity relationship database (admetSAR) [5], admetSAR 2.0 [5], and so on. In view of the importance of ADMET properties prediction, more useful methods should be developed and advanced so machine learning methods can be employed.

The Light Gradient Boosting Machine (LGBM) proposed by Ke et al. in 2017 [18] is a kind of advanced machine learning method. The LGBM method has many advantages in that the calculation speed is fast and can be applied to handle big data and provide better accuracy [19,20]. Due to these advantages, the LGBM is widely used by data scientists and has been applied in many fields [21,22,23,24]. To the best of our knowledge, up to now, there is only a report on the application of the LGBM method to the ADMET properties prediction for feature extraction purposes [25]. In light of the advantages of the LGBM method and its applications, the LGBM method was employed to establish models for the ADMET properties prediction of anti-breast cancer compounds. In order to further validate the performance of the LGBM models, the accuracy, precision, recall, and F1-score of the LGBM models were compared with those obtained from adaptive boosting (AdaBoost) and partial least squares-discriminant analysis (PLS-DA) models. The comparison results showed that the LGBM can establish more efficient ADMET properties prediction models. Therefore, there is a brighter prospect for the LGBM method in ADMET properties prediction for candidate compounds.

## 2. Data

The data applied in the presented study named “ADMET.xlsx,” “Molecular_Descriptor.xlsx” were obtained from the D-question of the 18th China Post-Graduate Mathematical Contest in Modeling (www.shumo.com/wiki/doku.php, accessed on 12 August 2021). At the same time, in order to make the model more robust, the public dataset downloaded from the website without any preprocessing has been used directly. In order to facilitate modeling, this research only considers the five given ADMET properties of compounds, namely:

(1) Intestinal epithelial cell permeability (Caco-2), which can measure the ability of compounds to be absorbed by the human body;

(2) Cytochrome P450 (CYP) 3A4 subtype (CYP3A4), which is the main metabolic enzyme in the human body, which can measure the metabolic stability of compounds;

(3) Human Ether-a-go-go Related Gene (hERG) can measure the cardiotoxicity of compounds;

(4) Human Oral Bioavailability (HOB) can measure the proportion of drug absorbed into the human blood circulation after entering the human body;

(5) The Micronucleus test (MN) is a method to detect whether a compound has genotoxicity.

The five ADMET properties of 1974 compounds are provided in “ADMET.xlsx”. The first column represents the SMILES (Simplified Molecular Input Line Entry System) formula for the structure of the compounds, and the following five columns correspond to the ADMET properties of each compound, providing the corresponding values by binary classification. For example:

① Caco-2: ‘1’ represents better permeability of intestinal epithelial cells, ‘0’ represents worse permeability of intestinal epithelial cells;

② CYP3A4: ‘1’ means that the compound can be metabolized by CYP3A4, ‘0’ means that the compound cannot be metabolized by CYP3A4;

③ hERG: ‘1’ means that the compound is cardiotoxic, ‘0’ means that the compound is not cardiotoxic;

④ HOB: ‘1’ represents a better oral bioavailability of the compound, ‘0’ represents a lower oral bioavailability of the compound;

⑤ MN: ‘1’ means the compound is genotoxic, ‘0’ means the compound is not genotoxic.

In the present study, the data in “Molecular_Descriptor.xlsx” have been used as x, and the data in “ADMET.xlsx” have been used as y. Then randomly select 75% of 1974 compounds data as the training set and 25% as the test set.

## 3. Theory

In this section, we give a brief introduction to the employed AdaBoost, PLS-DA, and LGBM methods used in this study.

### 3.1. PLS-DA

PLS-DA is one of the most well-known classification methods in chemometrics that was established on the basis of the PLS algorithm to replace the target variable value with a binary category variable [26]. The main advantage of the PLS-DA approach is its ability to handle highly collinear and noisy data [27]. The PLS-DA method has been applied in many fields and achieved successful results; these applications and a more detailed description of the PLS-DA method can be referred to in Refs. [28,29].

### 3.2. AdaBoost

AdaBoost [30] is a boosting-based method, which is a powerful machine-learning technique. A boosting algorithm is a method of combining weak classifiers to form strong classifiers. As for the AdaBoost algorithm, its core idea can be described as training different classifiers for the same training set and then combining the weak classifiers to form a stronger final classifier. The more detailed descriptions and applications of the AdaBoost method can be referred to in Refs. [31,32].

### 3.3. LGBM

LGBM was launched in 2017 [18]. It is a distributed and efficient framework for implementing the AdaBoost algorithm. It generates decision trees through the leaf-wise splitting method and searches for feature segmentation points through histogram-based algorithms. It aims to increase computational performance to solve big data prediction issues more effectively [33]. First of all, it generates a tree more complex than the level-wise splitting method using the leaf-wise splitting method, which can achieve higher accuracy. Secondly, it uses histogram-based algorithms to load continuous eigenvalues into discrete boxes in buckets and accelerates the calculation of the histogram of brother nodes through histogram difference, which can speed up the training process and achieve less memory consumption. Finally, it supports parallel learning, including feature parallelism and data parallelism. The more detailed descriptions and applications of the LGBM method can be referred to in Refs. [34,35].

### 3.4. Model Evaluation

Once a good model is obtained through the training set, the model’s performance will be evaluated by the test set using statistical parameters [36]. Generally speaking, accuracy is the most common measure of classification models. In order to decide the robustness and reliability of classification results sufficiently, some other parameters, such as precision, recall, and F1-score, also need to be integrated scientifically and comprehensively. As for accuracy, normally, the higher the accuracy, the better the classifier; precision represents the credibility of the model, recall measures the ability of the model to identify positive classes, and F1-score represents a comprehensive level of precision and recall of the established model [37]. In order to understand the concept of precision and recall, the terms such as true positive (TP), false positive (FP), true negative (TN), and false negative (FN) need to be defined first.

TP: An instance is a positive class and is predicted to be a positive class.

FP: An instance is a negative class and is predicted to be a positive class.

TN: An instance is a negative class and is predicted to be a negative class.

FN: An instance is a positive class and is predicted to be a negative class.

Accuracy refers to the proportion of correctly classified samples in the total samples among all discriminate results, and the general function is presented in Equation (1). The classification accuracy can be obviously presented in the form of a confusion matrix, which is illustrated in Figure 1.
(1)$$\mathrm{Accuracy}=\frac{\mathrm{TP}+\mathrm{TN}}{\mathrm{TP}+\mathrm{FP}+\mathrm{FN}+\mathrm{TN}}$$

Precision refers to how many samples a model identifies as being true and are a true positive; it does not care how many positive samples can be identified from the real positive; the general function can be presented as Equation (2).
(2)$$\mathrm{Precision}=\frac{\mathrm{TP}}{\mathrm{TP}+\mathrm{FP}}$$

Recall refers to how many of the actual positive examples are identified as positive by the model. The general function can be presented as Equation (3):(3)$$\mathrm{Recall}=\frac{\mathrm{TP}}{\mathrm{TP}+\mathrm{FN}}$$

The F1-score index combines the results of the Precision and Recall outputs with a value of 0–1, in which 1 represents the best output of the model and 0 represents the worst; the general function can be presented as Equation (4).
(4)$$F1\text{-}\mathrm{score}=\frac{2\times \left(\mathrm{Recall}\times \mathrm{Precision}\right)}{\mathrm{Recall}+\mathrm{Precision}}$$

## 4. Result and Discussion

In the present study, three models were established using PLS-DA, AdaBoost, and LGBM algorithms for classification prediction of the ADMET properties, including Caco-2, CYP3A4, hERG, HOB, and MN of the given compounds. All the calculations were implemented in Python 3.6.

For each property, the training (75% of all the compounds) and test set (the remaining 25% of all the compounds) were divided randomly, the classification prediction model was established based on the training set with dimensions of 1480 (the number of chemical compound samples) × 729 (the number of molecular descriptors), and the predictive ability of the established model was validated by the test set with dimensions of 494 (the number of chemical compound samples) × 729 (the number of molecular descriptors) [38].

In the process of randomly dividing the training set and test set, the random number seed controls the mode of dividing each time [39]. When the value of the random number seed is unchanged, the result of dividing is the same, and vice versa [40]. If this parameter is not given, the function will automatically select a random mode, and the results will be different. In the present study, in order to ensure the repeatability of the model, we selected the most common seed number, 1234, for the first time. At the same time, to ensure the robustness of the model, we randomly changed the seed number to repeat the modeling process twice; the seed number for the second and third random splitting were 5678 and 1278, respectively.

The specific analysis results were as follows:

### 4.1. Caco-2 Property

First, the classification prediction models for the ADMET properties of Caco-2 using the PLS-DA, AdaBoost, and LGBM methods based on different seed numbers were established, and the classification accuracy for the test set are summarized in Figure 2A. As can be seen from Figure 2A, the accuracies of the PLS-DA, AdaBoost, and LGBM models using different seed numbers were in the ranges 0.81–0.85, 0.88–0.90, and 0.90–0.91, respectively. Therefore, one can conclude that the selection of the seed number has different degrees of influence on the accuracy values, among which the PLS-DA models have the greatest influence and the LGBM models have the least influence, demonstrating that the LGBM models were the most robust. At the same time, we can see that the LGBM models have accuracy values higher than 0.90, indicating that the LGBM models can make better classification of the Caco-2 property, while the PLS-DA algorithm did not perform well [41].

For a more obvious presentation, the confusion matrices for the three models developed are compared and summarized in Figure 3. According to Figure 3, it can be seen that the LGBM models have the least miscalls (Figure 3C,F,I), which were lower than those obtained from AdaBoost models (Figure 3B,E,H) and PLS-DA models (Figure 3A,D,G).

In order to fully validate the performance of these established models, the indicators named precision, recall, and F1-score were also employed, and the calculated values are shown in Table 1. It can be seen that the LGBM models have higher precision, recall, and F1-score values than those of the PLS-DA and AdaBoost models. Based on the parameter values above, we can conclude that the LGBM model has higher accuracy and precision and is more robust for the classification of the Caco-2 property [42].

### 4.2. CYP3A4 Property

After the models were established for Caco-2, the classification prediction models for the ADMET properties of CYP3A4, using PLS-DA, AdaBoost, and LGBM methods based on different seed numbers, were established; the classification accuracy values are summarized in Figure 2B. As can be seen from Figure 2B, the accuracy values of the PLS-DA, AdaBoost, and LGBM models using different seed numbers were in the ranges 0.87–0.90, 0.93, and 0.94, respectively. In this case, the selection of the seed number had a small influence on the accuracies of the PLS-DA models and little impact on the AdaBoost and LGBM models, demonstrating that the AdaBoost and LGBM models were the same robustness and the LGBM models were more accurate with higher accuracy values.

Moreover, to facilitate the observation of the obtained results, the confusion matrices for the three developed models are summarized in Figure 4. According to Figure 4, we cannot only see that the established LGBM models have the highest accuracy but also have stable results in three repetitions.

Similarly, the precision, recall, and F1-score values of the three models are shown in Table 2. As one can see, the precision, recall, and F1-score values of the LGBM models established using the seed numbers 1234 and 5678 were all higher than those obtained from the PLS-DA and AdaBoost models. Though the precision value of the LGBM model established using the seed number 1278 was slightly lower than the other models, as far as the overall results were concerned, the LGBM models have better accuracy, precision, and stability [43].

### 4.3. hERG Property

With the same process as above, the classification accuracy values of the PLS-DA, AdaBoost, and LGBM models of hERG based on different seed numbers are summarized in Figure 2C. As can be seen from Figure 2C, the accuracies of the PLS-DA, AdaBoost, and LGBM models using different seed numbers were in the ranges 0.81–0.82, 0.86–0.89, and 0.89–0.91, respectively. One can conclude that the established LGBM models also have the best accuracy for the hERG property. Moreover, the confusion matrices for the three models developed are summarized in Figure 5. According to Figure 5, one can see that the accuracy values of the established models were not high, which may be caused by the imbalance of samples. However, the LGBM models still yielded the most satisfactory accuracy values. Similarly, the precision, recall, and F1-score of the established models are shown in Table 3. It can be seen that the LGBM models still show the best average performance with high precision, recall, and F1-score values [44].

### 4.4. HOB Property

Next, the classification prediction models for the ADMET properties of HOB using the PLS-DA, AdaBoost, and LGBM methods based on different seed numbers were also established; the classification accuracy values are summarized in Figure 2D. As can be seen from Figure 2D, the accuracies of the PLS-DA, AdaBoost, and LGBM models using different seed numbers were in the ranges of 0.79–0.80, 0.82–0.86, and 0.87–0.89, respectively. Moreover, the confusion matrices for the three models developed are summarized in Figure 6. In addition, the precision, recall, and F1-score of the three models are shown in Table 4. Not surprisingly, the performance of the LGBM models was more satisfactory compared with the PLD-DA and AdaBoost models.

### 4.5. MN Property

Finally, the models for the ADMET properties of MN using the PLS-DA, AdaBoost, and LGBM methods based on different seed numbers were also established; the accuracy values are summarized in Figure 2E. As can be seen from Figure 2E, the accuracies of the PLS-DA, AdaBoost, and LGBM models using different seed numbers were in the ranges 0.85–0.88, 0.94–0.96, and 0.95–0.97, respectively. Moreover, the confusion matrices for the three models developed are summarized in Figure 7. In addition, the precision, recall, and F1-score of the three models are shown in Table 5. The same conclusions can be obtained from the models above; the LGBM models have the best performance compared with the PLD-DA and AdaBoost models.

From the results of the LGBM modeling, it can be concluded that: (1) the LGBM algorithm showed better performance than the PLS-DA and AdaBoost algorithms in the process of establishing models to predict the molecular ADMET properties; (2) the performance of the LGBM models was more stable and less affected by sample changes.

Moreover, in the previous report, a new method named Two-Level Stacking Algorithm (TLSA) based on ensemble learning was proposed for ADMET classification for the same data set [45]. From the obtained results, we can find that the LGBM models perform better than TLSA for CYP3A4, hERG, and MN, while the results of the LGBM are not as good using TLSA for Caco-2 and HOB. In our opinion, one of the reasons for the better results obtained in the reference above may be that a feature selection process is included. The above research gives us inspiration; in a further study, we will do more in-depth research on the hybrid strategy by employing a feature selection method such as the genetic algorithm (GA) combined with the machine learning approach to obtain better performance.

## 5. Conclusions

In this study, two traditional machine learning models established using the PLS-DA and AdaBoost algorithms and a novel machine learning model established using the LGBM algorithm were applied to predict the ADMET properties named Caco-2, CYP3A4, hERG, HOB, and MN of anti-breast cancer compounds. The performance of the established models in the test set was evaluated using accuracy, precision, recall, and F1-score. The comparison of the results demonstrated that the LGBM models have higher accuracy and precision as well as more stable performance. It can be inferred that the LGBM can establish reliable models to predict the molecular ADMET properties and provide a useful tool for virtual screening and drug design researchers. In a further study, we will do more in-depth research to propose a hybrid strategy by employing the feature selection method combined with a machine learning approach to obtain better performance than a single machine learning method.

## Figures and Tables

**Figure 1 molecules-28-02326-f001:**
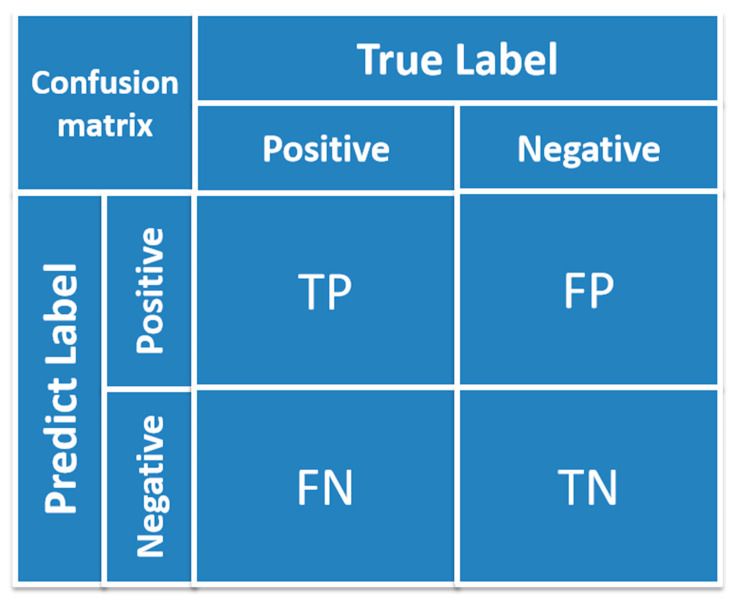
The presentation form of confusion matrix.

**Figure 2 molecules-28-02326-f002:**
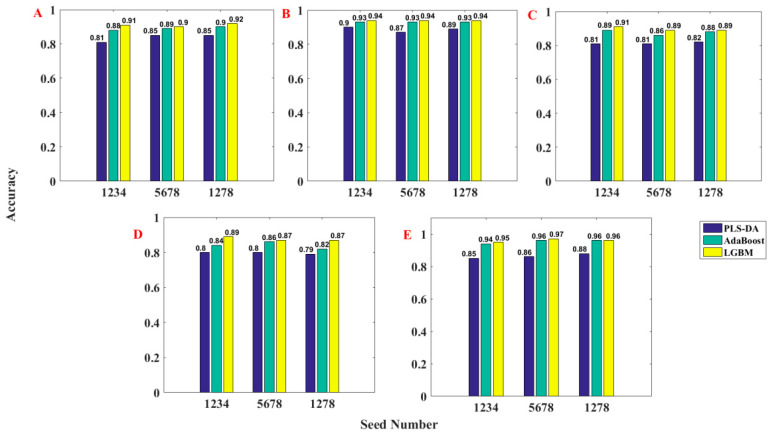
Accuracy histogram of the three models established using PLS-DA, AdaBoost, and LGBM method for Caco-2 (**A**), CYP3A4 (**B**), hERG (**C**), HOB (**D**), and MN (**E**).

**Figure 3 molecules-28-02326-f003:**
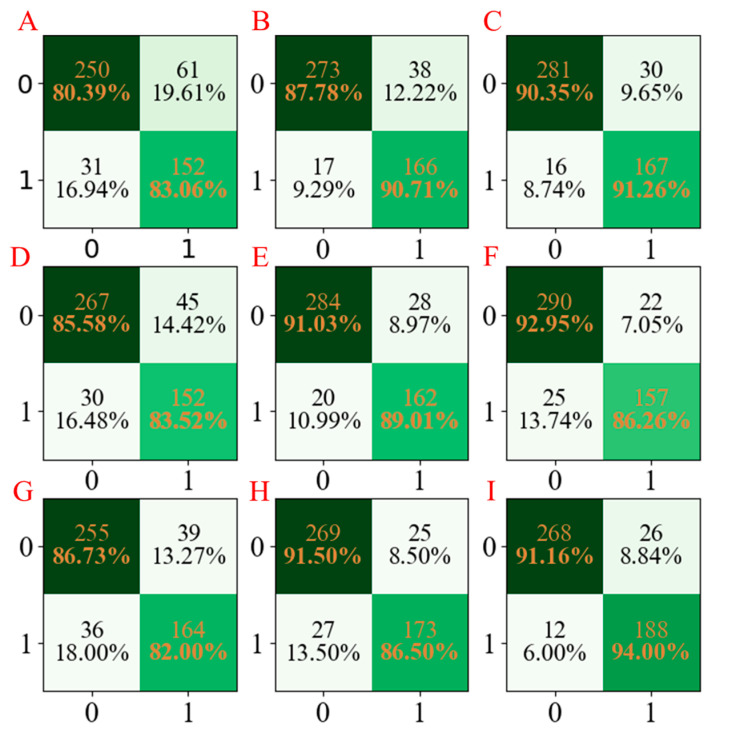
The confusion matrix of three models established for Caco-2 characteristics. The confusion matrix of the PLS-DA model using seed numbers 1234 (**A**), 5678 (**D**), and 1278 (**G**); the confusion matrix of the AdaBoost model using seed numbers 1234 (**B**), 5678 (**E**), and 1278 (**H**); the confusion matrix of the LGBM model using seed numbers 1234 (**C**), 5678 (**F**), and 1278 (**I**).

**Figure 4 molecules-28-02326-f004:**
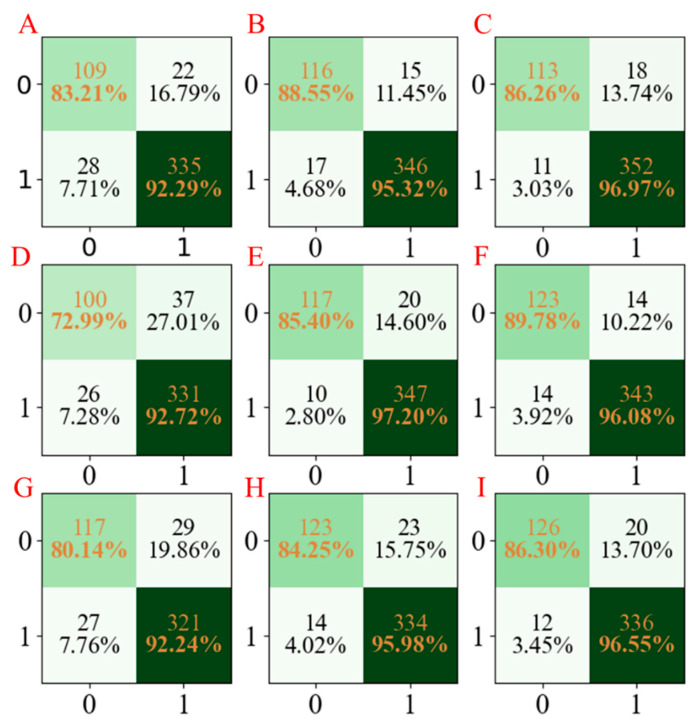
The confusion matrix of three models established for CYP3A4 characteristics. The confusion matrix of the PLS-DA model using seed numbers 1234 (**A**), 5678 (**D**), and 1278 (**G**); the confusion matrix of the AdaBoost model using seed numbers 1234 (**B**), 5678 (**E**), and 1278 (**H**); the confusion matrix of LGBM model using seed numbers 1234 (**C**), 5678 (**F**), and 1278 (**I**).

**Figure 5 molecules-28-02326-f005:**
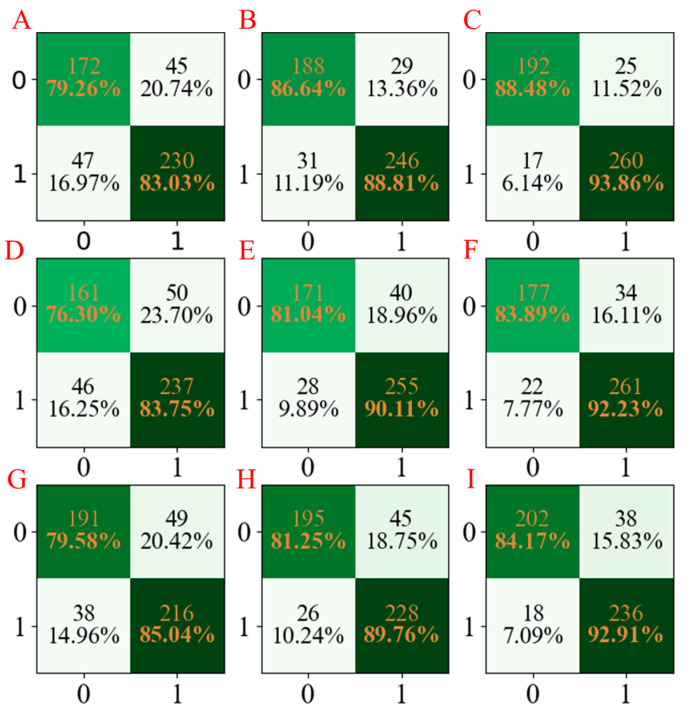
The confusion matrix of three models established for hERG characteristics. The confusion matrix of the PLS-DA model using seed numbers 1234 (**A**), 5678 (**D**), and 1278 (**G**); the confusion matrix of the AdaBoost model using seed numbers1234 (**B**), 5678 (**E**), and 1278 (**H**); the confusion matrix of the LGBM model using seed numbers 1234 (**C**), 5678 (**F**) and 1278 (**I**).

**Figure 6 molecules-28-02326-f006:**
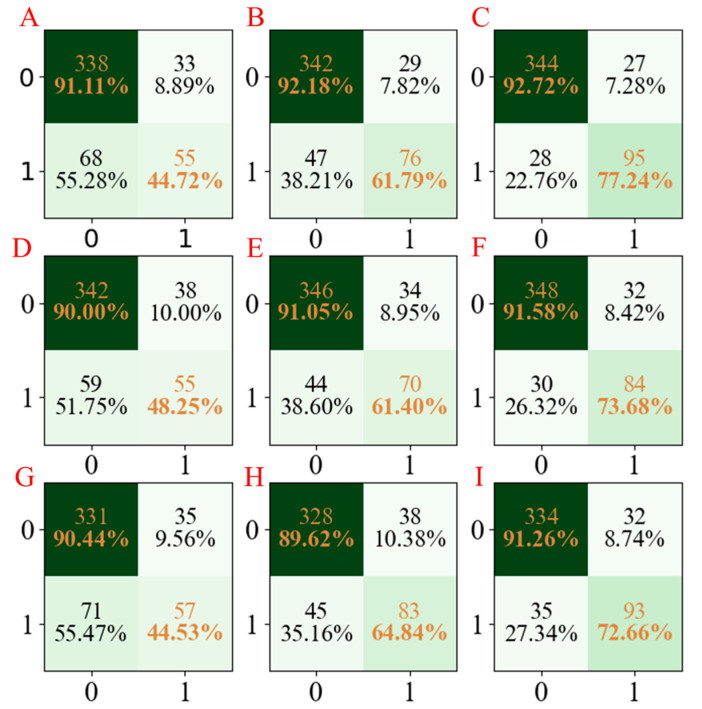
The confusion matrix of three models established for HOB characteristics. The confusion matrix of the PLS-DA model using seed numbers 1234 (**A**), 5678 (**D**), and 1278 (**G**); the confusion matrix of the AdaBoost model using seed numbers 1234 (**B**), 5678 (**E**) and 1278 (**H**); the confusion matrix of the LGBM model using seed number of 1234 (**C**), 5678 (**F**), and 1278 (**I**).

**Figure 7 molecules-28-02326-f007:**
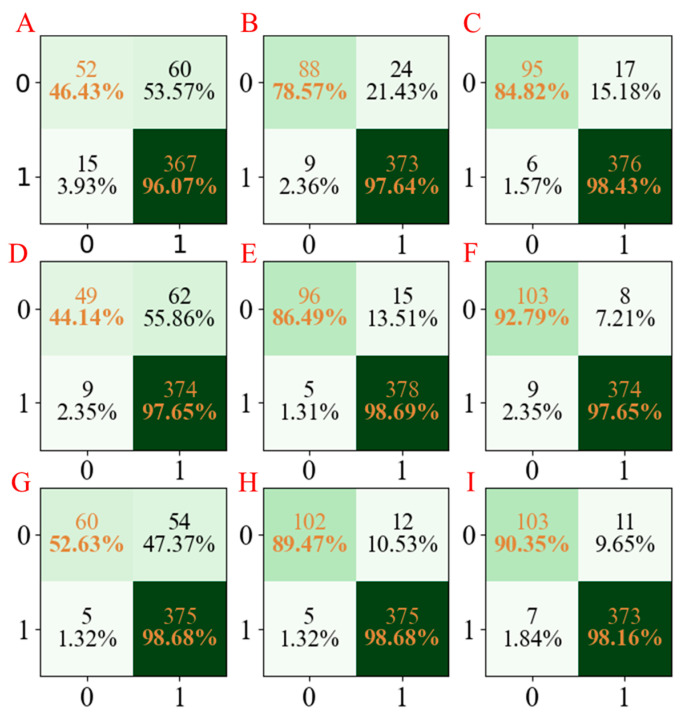
The confusion matrix of three models established for MN characteristics. The confusion matrix of the PLS-DA model using seed numbers1234 (**A**), 5678 (**D**), and 1278 (**G**); the confusion matrix of the AdaBoost model using seed numbers 1234 (**B**), 5678 (**E**), and 1278 (**H**); the confusion matrix of the LGBM model using seed numbers 1234 (**C**), 5678 (**F**), and 1278 (**I**).

**Table 1 molecules-28-02326-t001:** The precision, recall, and F1-score values of the established models for Caco-2 properties.

Methods	Seed Number	Precision	Recall	F1-Score
PLS-DA	1234	0.71	0.83	0.77
AdaBoost		0.80	0.89	0.84
LGBM		0.85	0.91	0.88
PLS-DA	5678	0.77	0.84	0.80
AdaBoost		0.86	0.85	0.85
LGBM		0.88	0.86	0.87
PLS-DA	1278	0.81	0.82	0.81
AdaBoost		0.88	0.88	0.88
LGBM		0.88	0.94	0.91

**Table 2 molecules-28-02326-t002:** The precision, recall, and F1-score values of the established models for CYP3A4 properties.

Methods	Seed Number	Precision	Recall	F1-Score
PLS-DA	1234	0.94	0.92	0.93
AdaBoost		0.94	0.96	0.95
LGBM		0.95	0.97	0.96
PLS-DA	5678	0.90	0.93	0.91
AdaBoost		0.95	0.96	0.95
LGBM		0.96	0.96	0.96
PLS-DA	1278	0.92	0.92	0.92
AdaBoost		0.95	0.96	0.95
LGBM		0.94	0.97	0.95

**Table 3 molecules-28-02326-t003:** The precision, recall, and F1-score values of the established models for hERG properties.

Methods	Seed Number	Precision	Recall	F1-Score
PLS-DA	1234	0.84	0.83	0.83
AdaBoost		0.90	0.90	0.90
LGBM		0.91	0.94	0.93
PLS-DA	5678	0.83	0.84	0.83
AdaBoost		0.88	0.88	0.88
LGBM		0.88	0.92	0.90
PLS-DA	1278	0.82	0.85	0.83
AdaBoost		0.87	0.91	0.89
LGBM		0.86	0.93	0.89

**Table 4 molecules-28-02326-t004:** The precision, recall, and F1-score values of the established models for HOB properties.

Methods	Seed Number	Precision	Recall	F1-Score
PLS-DA	1234	0.62	0.45	0.52
AdaBoost		0.69	0.63	0.66
LGBM		0.78	0.77	0.78
PLS-DA	5678	0.59	0.48	0.53
AdaBoost		0.69	0.70	0.70
LGBM		0.72	0.74	0.73
PLS-DA	1278	0.62	0.45	0.52
AdaBoost		0.68	0.60	0.64
LGBM		0.75	0.73	0.74

**Table 5 molecules-28-02326-t005:** The precision, recall, and F1-score values of the established models for MN properties.

Methods	Seed Number	Precision	Recall	F1-Score
PLS-DA	1234	0.60	0.96	0.91
AdaBoost		0.95	0.97	0.96
LGBM		0.96	0.98	0.97
PLS-DA	5678	0.86	0.96	0.91
AdaBoost		0.96	0.98	0.97
LGBM		0.98	0.98	0.98
PLS-DA	1278	0.87	0.99	0.93
AdaBoost		0.97	0.97	0.97
LGBM		0.97	0.98	0.98

## Data Availability

Not applicable.

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
