# Peer review of "Prediction of ADMET Properties of Anti-Breast Cancer Compounds Using Three Machine Learning Algorithms"

_molecules, 2023, doi:10.3390/molecules28052326_

Round 1

Reviewer 1 Report

The topic is interesting, and the author needs to clarify the below comments.

1. The numerical output is not disclosed in the abstract and the author needs to be appended.

2. The dataset of source or the repository link has not disclosed in the article, the author needs to upload the same.

3. Figure 1.  typos error needs to be corrected.

4. The confusion matrix provided in the Figure 3, 4, 5, 6 and 7 how the percentage has been used. In confusion matrix TP, TN, FP and FN alone can be used. With the help of these 4 parameters one can compute the accuracy, TPR, FPR etc., author needs to clarify the calculation made on these 2 tables. 

5. Author can update the algorithm in the article how the process are carried out.

Author Response

The topic is interesting, and the author needs to clarify the below comments.

Thank you for your acknowledgement and comments! We have revised our manuscript according to your suggestions.

  1. The numerical output is not disclosed in the abstract and the author needs to be appended.

Thank you for your suggestion. We have supplemented the precision values of the three algorithms in the summary in the revised manuscript.

  1. The dataset of source or the repository link has not disclosed in the article, the author needs to upload the same.

Thank you for your suggestion. The data set link has been added in the revised manuscript in Section 2.

  1. Figure 1. typos error needs to be corrected.

Thank you for your suggestion. We have carefully checked the spelling of the manuscript and revised Figure 1.

  1. The confusion matrix provided in the Figure 3, 4, 5, 6 and 7 how the percentage has been used. In confusion matrix TP, TN, FP and FN alone can be used. With the help of these 4 parameters one can compute the accuracy, TPR, FPR etc., author needs to clarify the calculation made on these 2 tables.

Thank you for your suggestion.TP, TN, FP and FN are used in the calculation of accuracy in formula. Accuracy refers to the proportion of correctly classified samples in the total samples among all discriminate results. In our manuscript, the accuracy values have been provided in the Figures 3-7.

  1. Author can update the algorithm in the article how the process are carried out.

Thank you for your suggestion. About the solution of the algorithm, we implemented it in Python 3.6 and only need to call the algorithm package to execute the algorithm. We have added corresponding information in the article.

Reviewer 2 Report

The study of Li et al. attempted to unravel established models to predict the ADMET properties of anti-breast cancer compounds, including Caco-2, CYP3A4, HERG, HOB, and MN. The authors are trying to solve this problem by application of machine learning models comprised of two traditional algorithms and a novel algorithm, which are PLS-DA, AdaBoost, and LGBM, respectively. The concept of this study looks good and reasonable, but several technical concerns need to be addressed, which are stated below.

1.       The format of the manuscript was not carefully checked. For example, the first three sections were not correctly numbered, consisting of the Introduction, Data, and Theory sections; Some texts were not centered in the columns in all tables.

2.       Some terminologies need to be corrected: “Then randomly select 75% of 1974 compounds data as the calibration set and 25% as the prediction set”. “The calibration set” and “the prediction set” need to be corrected as “the training set” and “the test set”. Correct all these related phrases in the manuscript.

3.       In lines 83-94, the authors mentioned that “this research only considers the five given ADMET properties of compounds…”, 5 properties did not represent for ADMET properties of anti-breast cancer compounds. The authors kindly explain why they chose these properties and need add those explanations and references to the manuscript.

4.       The paragraph in lines 95-106: “the following five columns correspond to the ADMET properties of each compound, providing the corresponding values by binary classification”. This study used binary classification to evaluate the ADMET properties, which should be accessed as the quantitative variables. The authors should provide the threshold of the “1” class and “0” class of these five properties.

5.       In the 3.4. Model evaluation, you should add the meaning of the indexes chosen to evaluate models rather than just mentioning the definition of these indexes.

6.       There is no comparison between the result of this study and other related research. The authors should read more publications and discuss them in the manuscript.

Author Response

The study of Li et al. attempted to unravel established models to predict the ADMET properties of anti-breast cancer compounds, including Caco-2, CYP3A4, HERG, HOB, and MN. The authors are trying to solve this problem by application of machine learning models comprised of two traditional algorithms and a novel algorithm, which are PLS-DA, AdaBoost, and LGBM, respectively. The concept of this study looks good and reasonable, but several technical concerns need to be addressed, which are stated below.

Thank you for your suggestion! We have revised the manuscript carefully according to your comments, and the responses to your comments are listed as follows one by one.

  1. The format of the manuscript was not carefully checked. For example, the first three sections were not correctly numbered, consisting of the Introduction, Data, and Theory sections; Some texts were not centered in the columns in all tables.

Thank you for your suggestion. We have carefully revised the number and icon in the manuscript.

  1. Some terminologies need to be corrected: “Then randomly select 75% of 1974 compounds data as the calibration set and 25% as the prediction set”. “The calibration set” and “the prediction set” need to be corrected as “the training set” and “the test set”. Correct all these related phrases in the manuscript.

Thank you for your suggestion. We have changed all related phrase in the manuscript.

  1. In lines 83-94, the authors mentioned that “this research only considers the five given ADMET properties of compounds…”, 5 properties did not represent for ADMET properties of anti-breast cancer compounds. The authors kindly explain why they chose these properties and need add those explanations and references to the manuscript.

Thank you for your suggestion. In our study, a public data set has been used, and this public data set given these five properties. The established model can be used to predict the given five properties, and to some extent indicates the feasibility of the models to predict other properties, thus we did not consider using other ADMET properties of anti-breast cancer compounds. We have added corresponding descriptions to the manuscript.

  1. The paragraph in lines 95-106: “the following five columns correspond to the ADMET properties of each compound, providing the corresponding values by binary classification”. This study used binary classification to evaluate the ADMET properties, which should be accessed as the quantitative variables. The authors should provide the threshold of the “1” class and “0” class of these five properties.

Thank you for your suggestion. In the public data set, the “1” class and “0” class of these five properties have been given explicitly, so that we do not need to judge according to the threshold. Here, the symbol descriptions for the given ADMET properties are listed in following table (Table 1).

Table 1 The symbol descriptions for the given ADMET properties

Properties

Symbolic

Symbolic meaning

Caco-2

1

Better permeability of intestinal epithelial cells

0

Worse permeability of intestinal epithelial cells

CYP3A4

1

The compound can be metabolized by CYP3A4

0

The compound cannot be metabolized by CYP3A4

hERG

1

The compound is cardiotoxic

0

The compound is not cardiotoxic

HOB

1

Better oral bioavailability of the compound

0

Lower oral bioavailability of the compound

MN

1

The compound is genotoxic

0

The compound is not genotoxic

  1. In the 3.4. Model evaluation, you should add the meaning of the indexes chosen to evaluate models rather than just mentioning the definition of these indexes.

Thank you for your suggestion.

The accuracy, precision, recall and F1-score are integrated scientifically and comprehensively to decide the accuracy, robustness and reliability of classification results. As for accuracy, normally, the higher the accuracy, the better the classifier, precision represents the credibility of the model, recall measures the ability of the model to identify positive classes, and F1-score represents a comprehensive level of precision and recall of the established model. We have added corresponding descriptions about the meaning of the indexes in the revised manuscript. We hope that such revision is appropriate.

  1. There is no comparison between the result of this study and other related research. The authors should read more publications and discuss them in the manuscript.

Thank you for your suggestion!

In the previous report, a new method named Two-Level Stacking Algorithm (TLSA) based on ensemble learning was proposed for ADMET classification for the same data set. From the obtained results, we can find that the LGBM models perform better that TLSA for CYP3A4, hERG, and MN, while the results of LGBM are not as good as these of TLSA for Caco-2 and HOB. In our opinion, one of the reasons for the better results obtained in the above reference may be that a feature selection process is included. The above research gives us an inspiration, in the further study, we will do more in-depth research on the hybrid strategy by employing feature selection method combined with machine learning approach to obtain better performance. We have supplemented relevant discussion in Section 4 in the revised manuscript.

Round 2

Reviewer 1 Report

This revision has done well, author needs to revise the following.

1. The reference of the dataset needs to be cited in reference section.

2. In Fig 3 A, author has mentioned TP as 152 sample with 83.06% and TN as 250 sample with 80.39%, can author explain how the percentage has received for Fig A to I.

Reviewer 2 Report

Authors have significantly revised the manuscript as per my comments, therefore, I would like to recommend this MS for publication in Molecules.